# Surface-Step-Induced Magnetic Anisotropy in Epitaxial LSMO Deposited on Engineered STO Surfaces

**DOI:** 10.3390/ma13184148

**Published:** 2020-09-17

**Authors:** Jakub Pawlak, Antoni Żywczak, Jarosław Kanak, Marek Przybylski

**Affiliations:** 1Faculty of Physics and Applied Computer Science, AGH University of Science and Technology, 30-059 Kraków, Poland; marprzyb@agh.edu.pl; 2Academic Centre for Materials and Nanotechnology, AGH University of Science and Technology, 30-059 Kraków, Poland; zywczak@agh.edu.pl; 3Department of Electronics, Faculty of Informatics, Electronics and Telecommunications, AGH University of Science and Technology, 30-059 Kraków, Poland; kanak@agh.edu.pl

**Keywords:** magnetic anisotropy, interface engineering, perovskite, LSMO, STO

## Abstract

Changes in stoichiometry, temperature, strain and other parameters dramatically alter properties of LSMO perovskite. Thus, the sensitivity of LSMO may enable control of the magnetic properties of the film. This work demonstrates the capabilities of interface engineering to achieve the desired effects. Three methods of preparing STO substrates were conducted, i.e., using acid, buffer solution, and deionized water. The occurrence of terraces and their morphology depend on the preparation treatment. Terraces propagate on deposited layers and influence LSMO properties. The measurements show that anisotropy depends on the roughness of the substrate, the method of preparing the substrate, and oxygen treatment. The collected results suggest that the dipolar mechanism may be the source of LSMO anisotropy.

## 1. Introduction

The latest spintronics research heavily exploits oxide materials [1,2,3,4,5,6,7], in which perovskites play a unique role due to their special properties [8,9,10]. One of the widely used materials is La_0.67_Sr_0.33_MnO_3_ (LSMO) [11,12,13,14,15,16], which is characterized by a Curie temperature above room temperature (RT) and nearly 100% spin polarization of electrons at the Fermi level [17]. For this reason, LSMO is often used as a ferromagnetic electrode in a multiferroic tunnel junction. Depending on strain, temperature and the ratio of lanthanum to strontium atoms, LSMO can be a conductive ferromagnetic, antiferromagnetic, or a paramagnetic isolator [18,19,20].

An ideal substrate for the LSMO layer is monocrystalline perovskite SrTiO_3_ (STO). Its structural complementarity with LSMO (the same structure, i.e., perovskite, and similar lattice constant) allows epitaxial growth of a multilayer when the pulsed laser deposition (PLD) method is applied [12,13]. Therefore, perovskite materials, such as LSMO, BaTiO_3_ [21,22,23] and STO [24,25], have recently been extensively tested for tunnel junction applications [26,27,28,29,30].

Currently, there are several unresolved problems of a fundamental nature that at least hinder the formation of multiferroic junctions with parameters allowing for their practical applications. One of these problems is too low magnetocrystalline anisotropy of LSMO layers, resulting in a lack of two distinctly different magnetization configurations/settings in one of the electrodes. There are also other sources of anisotropy, e.g., magnetoelastic [31,32,33]. Currently, efforts are being made to induce an additional direction of easy magnetization in magnetic layers using nanostructuring as a source of a change in shape anisotropy [34,35]. Magnetic layers could exhibit magnetic anisotropy because of the introduction of disturbance sources of the atomic arrangement symmetry, which can lead to additional contribution to the anisotropy energies, like steps on the STO substrate (step anisotropy) [36].

The aim of this article is to show how to control the magnetic anisotropy of LSMO. It can be done by introducing disturbance sources of the atomic arrangement symmetry, such as steps on the STO substrate and thus on the deposited LSMO layer. Different STO substrate preparation methods lead to different roughnesses of the surface, and thus different magnetic anisotropies. Magnetic anisotropy also depends on the amount of oxygen during and after deposition.

## 2. Materials and Methods

Three preparation methods of a single-crystal STO (100) substrate were conducted. The selected substrates showed a similar miscut effect (measured using the XRD measurement). The miscut is the effect of a slight mismatch between the polishing direction and the atomic plane. It results in a stepped surface with a random terrace direction and the width of the terraces corresponding to the miscut angle. The first method involves successive ultrasonic rinses in HCl-HNO_3_ (3:1), deionized water, acetone and methanol, followed by annealing at 1000 °C in an oxygen atmosphere for an hour [37]. For simplicity, let us call this method “acid”. The second method involves successive rinses in buffer solution NH_4_F: HF (7:1), acetone, methanol and deionized water, followed by annealing at 1000 °C in an oxygen atmosphere for an hour [38], hereafter referred to as the “buffer” method. The third one is based only on ultrasonic bath in deionized water and annealing in an oxygen atmosphere at 1000 °C for 1 h [39], called the “DIW” (deionized water) method.

Thin LSMO films were grown onto the prepared STO by the pulsed laser deposition (PLD) technique, using a 248 nm excimer laser system (Coherent COMPex Pro 110F, Santa Clara, CA, USA) operated at an energy density of ~2 J·cm^−2^, with the distance between the sample and the target equal to 6.5 cm, pulse width of 20 ns, repetition rate of 10 Hz, and 1200 as the number of pulses. The aim was to obtain stoichiometric LSMO. Laser deposition was performed under 200 mTorr partial oxygen pressure and a substrate temperature of 750 °C. The thicknesses of the deposited LSMO layers are about 20 nm.

The surface morphology and the grain size dependence on the buffer were examined with an atomic force microscope (AFM) using the NTEGRA Aura—NT-MDT system (Amsterdam, The Netherlands). The crystallographic structure of the samples was investigated by x-ray diffraction (XRD) measurements, using an X’Pert MPD diffractometer (Malvern, UK). The hysteresis loops were measured by a vibrating sample magnetometer (VSM) LakeShore 7407 (Westerville, OH, USA) at room temperature in an external magnetic field applied at different angles in the plane of the sample.

## 3. Results and Discussion

### 3.1. Preparation and Roughness

The proper preparation of the STO substrate is crucial for obtaining high quality magnetic properties of the deposited layers. The substrate must have sufficient smoothness to allow epitaxial growth. Three different substrate preparation methods were tested, followed by atomic force microscopy (AFM) measurements. Table 1 shows the roughness of the differently prepared samples.

Figure 1 shows the surface of the STO samples examined with AFM after appropriate chemical treatment, i.e., acid, buffer and DIW method. Visible terraces are the result of the existence of the miscut and the appropriate treatment that affects the nature of the terraces. Figure 1b shows that the buffer method results in a more wavy surface than the acid method (darker and lighter regions). As can be seen in Figure 1c, the preparation process itself reveals additional grooves, visible as darker stripes. This is one of the effects that cause increased roughness.

The roughness is calculated from the height profile of the samples measured perpendicular to the terraces. It is calculated as the average absolute value of the height. In the ideal case, the maximum value would be half the unit cell (STO 3.89 Å, LSMO 3.87 Å), and the roughness would be a quarter, i.e., 0.97 Å. The measured roughness for all samples is less than half of the height of the unit cell, which means that they all have an atomically smooth surface. However, the roughness at this level should be understood as a higher number of defects, which will directly affect the deposited layers on such a substrate. In the real case, it is often difficult to distinguish where the terrace ends and where it begins, and the model terrace rarely takes place.

Figure 1 shows the differences between the height profiles of the STO samples prepared with the “acid”, “buffer” and “DIW” methods. The “acid” method results in a mostly uniform and smooth surface, but often with a half unit-cell height of terraces. The “DIW” method often results in terraces with one and a half height of the unit-cell steps (areas with SrO and TiO_2_ termination), whereas the “buffer” method results in a surface rougher than that obtained with the “acid” one but smoother than in the case of the “DIW” method. Sometimes even larger defects appear. Generally, a good measure of these effects is roughness, which even for clearly stepped surfaces may have different values (Table 1). The smoothest surface is the result of the most invasive preparation, such as the use of acids and buffer solutions. A rougher substrate can be obtained by only annealing and ultrasonic bath in deionized water.

### 3.2. Preparation and Roughness

The existence of terraces on the deposited layers can be observed. They have the same width and direction, i.e., the surface of the deposited LSMO reproduces terraces of the STO surface.

This effect is visible even in the case of several dozen nm of the sputtered layers. The AFM measurements of the STO sample before and after the deposition of LSMO (prepared using the “buffer method”) are shown in Figure 2. A similar effect was also observed for the other samples. Terrace propagation suggests layer-by-layer growth, and every layer of the deposited material has a terraced surface. This effect can be exploited to obtain a terraced surface with a certain roughness on different materials deposited on STO in a multilayer structure.

### 3.3. Step-Induced Anisotropy

Figure 3a shows the offset angle of the LSMO/STO measured for different directions in the sample plane. The offset is the angle between crystalline planes and the sample plane. The orientation of the crystalline planes was determined using rocking curve measurements at STO (002) peak. The exact orientation of the sample plane was checked by tilting of the sample at incidence beam parallel to the sample plane and detector placed in front of an x-ray source. The zero offset occurs for the direction along the terrace (no inclination). The maximum value of the offset is the miscut angle. Figure 3b shows the correlation between the steps and the magnetic easy and hard axes. The hysteresis loops of the LSMO/STO samples were measured by VSM for the field in the plane of the sample. The LSMO samples show the existence of magnetic anisotropy in the plane of the sample (Figure 3c). There is a correlation between the direction of the terraces (resulting from the miscut) and the easy and hard axes. The magnetic hard axis occurs in the direction perpendicular to the terraces, whereas the easy axis along the terraces (Figure 3b). Please note that the easy axis from the VSM measurement corresponds to the zero point of the offset (obtained from the XRD measurement) and the direction along the terraces obtained from the AFM measurements. Similar results and relationships were obtained for the other samples, as well. All measured samples have similar terrace widths; thus, the effect of the terraces density on anisotropy was not observed in our samples. No correlation was observed between the anisotropies and the crystallographic directions as well.

Thin ferromagnetic films grown on stepped surfaces show complex magnetic anisotropy due to their reduced symmetry [40,41]. The easy direction of magnetization is usually along the step edges, but it may switch to the plane perpendicular to the steps for films of a few monolayers in thickness. Such a magnetization orientation has been found experimentally using the magneto-optical Kerr effect for stepped Fe and Co ultrathin films with well-defined structures, grown epitaxially on Ag and Cu vicinal substrates and optionally covered by Au and Cu [42,43]. The observed orientation and canting of magnetization have been explained within a theoretical model by an interplay of the shape anisotropy due to magnetic dipolar interaction and three magnetocrystalline anisotropy terms arising from the spin–orbit interaction for films with stepped surfaces. The magnetic anisotropy energy has been found to include both a surface term and a volume contribution, which presumably results from the structural distortion of the film structure above the step edges of the underlying substrate due to the lattice mismatch between the film and the substrate.

Figure 4 shows the XRD measurement of the STO substrate prepared by the “buffer” method and the same substrate after LSMO deposition. It shows crystalline growth of the LSMO layer. The presented XRD pattern shows not only high crystallinity of the sample, but also demonstrates high quality of interfaces—one can note the appearance of Laue oscillations in Figure 4b. Similar results were obtained for the other samples.

### 3.4. Roughness and Anisotropy

Figure 5a shows the hysteresis loops measured perpendicular to the terraces (along the hard axis) for substrates prepared using various methods. As it turns out, different methods of substrate preparation have an impact on the anisotropy field. The reason for these differences is the roughness of the substrates, which induces a relationship between roughness and anisotropy field (Figure 5b). In this article, we use the anisotropy field as a measure of anisotropy, i.e., the magnetic field that must be applied along the hard axis to drag the direction of magnetization of the sample along the external field. It was measured using VSM at room temperature. It should be added that the hysteresis loops measured along the easy axis are rectangular for all samples. The dependence of the anisotropy field on roughness can be explained by the dipolar mechanism proposed by Arias and Mills [44]. It occurs when roughness has a uniaxial character.

In the lowest order of the perturbation theory, we have a simple formula for approximation [44]:(1)$$\frac{\Delta E}{\Delta E^{\left(0\right)}}\approx \frac{2\alpha }{\alpha +Q}$$
where Q—reciprocal width of the Neel-type domain wall, α—miscut angle, $\Delta E^{\left(0\right)}$—total energy of the system. Using the expression for the effective magnetic field $\Delta E=MDH_{A}$, where *M*—magnetization, *D*—thickness of the layer, $H_{A}$—anisotropy field, we can formulate the following relation [44]:(2)$$H_{A}\sim \frac{\alpha }{\alpha +Q}$$

In their model, Arias and Mills assumed the ideal form of terraces. As shown in Figure 1, this is not always the case. Instead of an angle, we use roughness as a measure of the source of the dipolar mechanism. In our case, due to the same width of the terraces of all samples, it will be a good approximation as α≈σ/L, where σ is the roughness and *L* is the average width of the terrace.

Thus, we get:(3)$$H_{A}\sim \frac{\sigma }{\sigma +Q}$$

For our sample, we are in the regime of low roughness, so we should see a simple linear dependence of magnetic anisotropy as a function of roughness
(4)$$H_{A}\sim \frac{\sigma }{Q}\sim \sigma$$

Figure 5b shows the measured anisotropy versus the roughness for the LSMO/STO samples as a clearly linear dependence. It can also be concluded that the use of a specific method of STO preparation makes it possible to control the roughness of the substrate. Thus, it leads to different anisotropy fields of the deposited layers.

### 3.5. Anisotropy and Oxygen

All the previously shown results concerned quickly cooled (after deposition) LSMO samples without adding additional oxygen after deposition. Figure 6a presents the hysteresis loops of three different LSMO/STO samples measured along the hard axis: (1) “Low oxygen”: a quickly cooled sample after deposition at 200 mTorr oxygen pressure, (2) “Annealed”: the same sample after additional annealing in an oxygen atmosphere and at 800 °C for 1 h, and (3) “High oxygen”: an LSMO sample slowly cooled after deposition with continuous addition of oxygen up to the value of 50 Torr. The measurement shows that anisotropy depends not only on the roughness and method of substrate preparation but also on oxygen treatment. LSMO is very sensitive to various types of influences and particularly to oxygen treatment due to its rich phase diagram and a Curie temperature close to room temperature. Figure 6b shows the saturation magnetization for different types of oxygen treatment during and after deposition. This phenomenon is related to oxygen vacancies, which are known to affect magnetic properties in manganite films [45]. The oxygen treatment leads to an increase in the concentration of Mn^3+^ ions because of missing O-Mn bonds. The oxygen vacancy thus leads to the destruction of Mn^3+^–Mn^4+^ chains, in which hopping electrons in the double exchange mechanism lead to ferromagnetic ordering. In turn, the Mn^3+^–Mn^3+^ ion chains in the superexchange mechanism may lead to segregation into a form of larger antiferromagnetic clusters [46]. This results in a degradation of the ferromagnetic properties. Oxygen deficiency also leads to point defects that become centers of the pining domain wall in LSMO. Both these mechanisms cause an increase in the magnetic anisotropy (anisotropy field—Figure 6c) [45]. Oxygen depletion can be particularly strong in interface areas, where by nature most defects occur. It may cause the dipolar mechanism due to changes in the effective surface charge density.

## 4. Conclusions

Three methods of STO substrate preparation were tested to obtain atomically flat surfaces. All these methods resulted in visible surface terraces with heights of about a unit cell and a roughness of less than 2 Å. LSMO layers were deposited on engineered STO substrates using PLD method. The propagation of the terraces on the deposited layers, with the same width and direction as those on the substrate, was confirmed even in the case of several dozen nm of LSMO. The use of appropriate deposition conditions (oxygen treatments and annealing) and the existence of terraces causes defects and oxygen vacancies, and thus magnetic anisotropy. The existing correlations between terraces and magnetic anisotropy have been confirmed in the samples by VSM, XRD and AFM measurements. The magnetic hard axis is perpendicular to the terraces and the magnetic easy axis is along the terraces. The correlation between the preparation of the substrate and its roughness as well as between its roughness and magnetic anisotropy has been shown. The most invasive methods, such as the use of acids and buffers, give the smoothest surface and lowest magnetic anisotropy. The highest anisotropy occurs for rougher surfaces obtained, for example, by only annealing and ultrasonic bath in deionized water. The dependence of the anisotropy field on roughness can be explained by the dipolar mechanism proposed by Arias and Mills [44]. It occurs when roughness has a uniaxial character. In this case, the magnetic energy per volume is proportional to the roughness measured perpendicular to the terraces.

## Figures and Tables

**Figure 1 materials-13-04148-f001:**
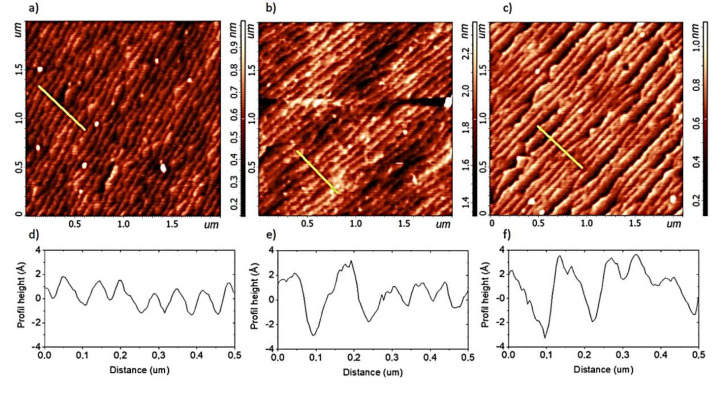
The AFM measurements of the surfaces of the STO samples prepared with different methods (**a**) acid, (**b**) buffer, (**c**) DIW, with the corresponding height profiles measured perpendicular to the terraces (**d**–**f**) marked as the yellow line on the AFM graph.

**Figure 2 materials-13-04148-f002:**
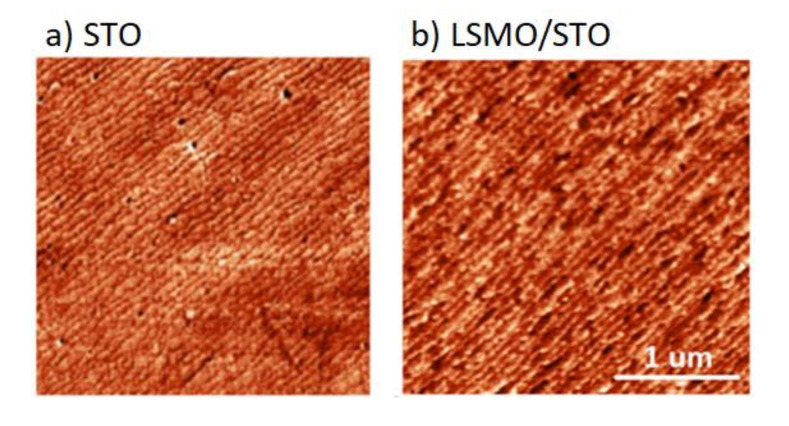
AFM measurement of (**a**) the prepared substrate (using the “buffer” method), and (**b**) after the deposition of LSMO using the PLD technique.

**Figure 3 materials-13-04148-f003:**
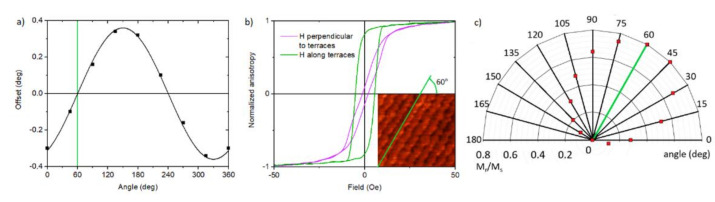
The measurements of the LSMO/STO sample prepared with the “buffer” method showing the correlation between (**a**) the STO (002) offset values as a function of the different directions in the sample plane (**b**) the magnetic hysteresis loop measured along the easy (green) and hard (pink) axes. In the inlet the AFM measurement of the sample surface, (**c**) the proportion of remanence to the magnetic saturation for different angles between the external magnetic field H and the [100] direction.

**Figure 4 materials-13-04148-f004:**
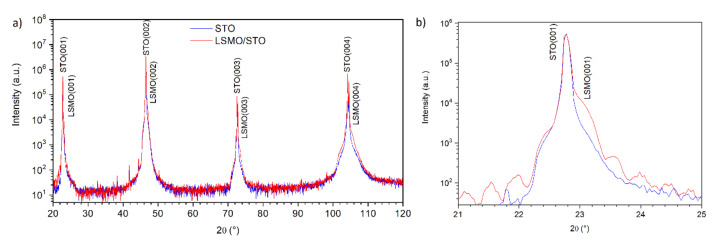
(**a**) The XRD measurement of the STO substrate prepared by “buffer” method (blue) and the same sample after LSMO deposition (red). (**b**) Close-up on (100) reflex details.

**Figure 5 materials-13-04148-f005:**
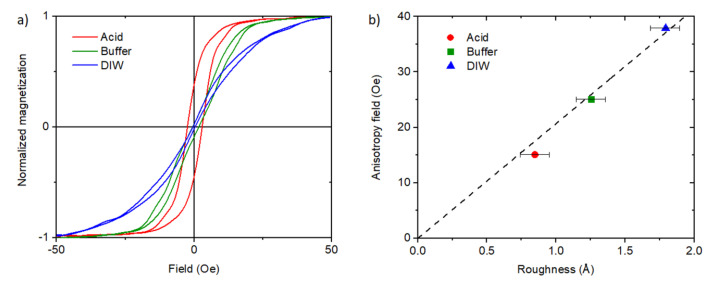
(**a**) The hysteresis loops measured by VSM with the magnetic field perpendicular to the terraces (hard axis) for the samples prepared using various methods. (**b**) The dependence of the anisotropy field (measured for the hard axis) on the roughness with the linear fit to the Arias and Mills dipolar mechanism model. The methods of sample preparation mentioned in the Experimental section are marked as red circle—“acid”, green square—“buffer”, and blue triangle—“DIW”.

**Figure 6 materials-13-04148-f006:**
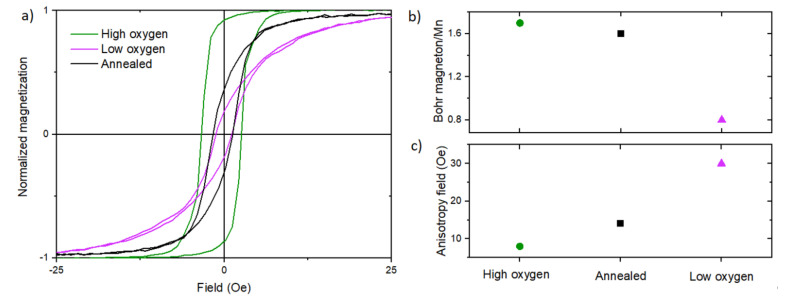
(**a**) The magnetic hysteresis loops measured by VSM along the hard axis of: a rapidly cooled LSMO/STO sample after sputtering (pink), the same sample after 1 h oxygen annealing at 800 °C (black), and a slowly cooled LSMO/STO sample with simultaneous continuous addition of oxygen up to the value of 50 Torr (green). (**b**) The saturation magnetization of LSMO per manganese atom and the anisotropy field of a rapidly cooled LSMO/STO sample after sputtering (pink triangle), the same sample after 1 h oxygen annealing at 800 °C (black square), and a slowly cooled LSMO/STO sample with simultaneous continuous addition of oxygen up to the value of 50 Torr (green circle). (**c**) The anisotropy field of abovementioned samples.

**Table 1 materials-13-04148-t001:** The roughness of the differently prepared samples measured by AFM.

Method	STO	STO/LSMO
Acid	0.8 Å	0.9 Å
Buffer	1.3 Å	1.3 Å
DIW	1.8 Å	1.8 Å

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
