# Peer review of "Surface-Step-Induced Magnetic Anisotropy in Epitaxial LSMO Deposited on Engineered STO Surfaces"

_materials, 2020, doi:10.3390/ma13184148_

Round 1

Reviewer 1 Report

Presented paper describe actual and important topic of the interface induced magnetic anisotropy in thin epitaxial films, namely appearance of in-plane uniaxial anisotropy in LSMO thin films grown on STO (001) surface with a miscut angle. However, authors miss some key points both in experiment description and result discussion. Thus, this work can be published only the major revision.

Here is the list of the controversial points and comments:

  1. Authors do not provide in the text any discussion on the miscut angle of the used STO (001) wafers. As a rule, wafer manufacturers indicate only the range of permissible miscut, but not the specific miscut angle itself. Did all used substrates have the same misorientation?
  2. It would be appropriate if authors describe the effect and necessity of each specific chemical treatment method in 2. Materials and Methods section. I.e. it is known that buffered HF treatment is commonly used to obtain thermally stable TiO2-terminated STO [doi:10.1063/1.1771461]. In addition, choosing of substrate annealing procedure is not discussed in the text. One can assume that the annealing regime neither that chemical treatment can have a decisive effect on the wafer surface roughness. For e.g. step bunching can occurs at high temperatures.
  3. It is worth to note target to substrate distance during PLD process in the 2. Materials and Methods section.
  4. Authors do not provide in the text the thickness of the deposited LSMO film. One can suggest that magnetically dead layers can appears at the interface regions (doi:10.1063/1.5005913). Have studies been carried out how both uniaxial in-plane magnetic anisotropy or saturation magnetization depend on the film thickness?
  5. Concerning the crystalline quality of the grown films – authors do not provide any XRD or RHEED data proving its crystal structure and stoichiometry. It is known that due to orthorhombic distortion twinned domain structure can appears in LSMO films (doi: /10.1088/1367-2630) which can also alter it magnetic anisotropy.
  6. Lines 81-82 describing the miscut effect on surface morphology can be moved in Introduction section.
  7. Both description of the experiment and results of the “low angle XRD measurements” presented on the line 122-123 are completely not clear. Which parameter was measured and how? What does term “offset” means? Additional information must be provided.
  8. Line 236 (Conclusions). Citing: “The propagation of the terraces on the deposited layers, with the same width and direction as those on the substrate, was confirmed even in the case of several dozen nm and various materials used.” That term “material” means – as I understand authors report only on the LSMO growth?
  9. I doubt about Link 43 (line 296), if authors mean [doi:10.1103/PhysRevB.60.7395] – it seems to be incorrectly written.
  10. Additional works describing the effect of the stray field induced uniaxial anisotropy in the ferromagnetic films grown on stepped surfaces show can be sited – i.e. [https://www.nature.com/articles/srep02148]
  11. Fig 1 is not fit in page width.

Reviewer 2 Report

The authors describe a study of the magnetic anisotropy by introducing terraces into STO substrates before depositing LSMO on top of it by PLD. Additional experiments must be performed before acceptance can be granted.

  • Most severly, the authors miss describing a control experiment of an untreated substrate of STO and the properties of a corresponding LSMO film. If the surface effects induced by the suggested pretreatments are real, then the authors should show that the results clearly distinguish from an untreated substrated. I don't think that the DIW sample must necessarily serve as such a blank state.
  • Also, XRD results of the films should be presented. The authors use in one case very strong fluorination conditions. Is it to be ruled out that fluoride is present at the surface of the film after the treatment, and that this might also impact film growth (though F does not incorporate into STO films easily, DOI: 10.3390/ma11071204). XPS could give information on this aspect, and the authors should comment on this.
  • The references might need be reformatted.

Reviewer 3 Report

Authors of reviewed manuscript studied the influence of STO substrate roughness and the LSMO films stoichiometry on the magnetic anisotropy of films. This topic is relevant and worthy of study.The obtained resultscan bepublished inthis journal after the major revisions.

The main comments are the following:

  1. Substrate STO with orientation (001) has a fourth axis of symmetry, therefore, terraces with a second axis of symmetry are possible only if the substrate is cut at some angle to some crystallographic direction. However, the authors do not emphasize this.
  2. In addition, the nature of the formation of these terraces is unclear. The authors must explain their nature and what caused them - is it an etching product or natural atomic steps or something else.
  3. It is unclear how the authors determined the anisotropy shown in Fig. 4a and 5c.
  4. It is also not very clear what is shown in Fig. 3a.
  5. Authors of manuscript does not specified thickness of the grown layer.
  6. Moreover, the authors argue that the growth of the films occurs by a layer-by-layer mechanism, and the anisotropy of magnetization is caused by steps on the substrate surface only based on atomic force microscope data. To prove this, it would be necessary to investigate the film with a transmission electron microscope, since another nature of anisotropy is also possible. One of them may be the mosaic structure of the film.

Round 2

Reviewer 1 Report

Thanks for your reply and corrections. In my opinion, there are a few small points that require minor corrections.

Here is the list of my thoughts:

1) Presented XRD pattern shows not only high crystallity of the sample (to be honest, symmetrical XRD Th-2Th scans gives do not give full infopmation), but also demonstrate high quality of interfaces - one can note the appearance of Laue oscillations in Fig4 b). This aspect can be noted in text if the authors deem it necessary. Film thickness can be also judged from the period of oscillations. 

2) The details of grazing incidence XRD experement are still not clear for me - that was measured - direct beam?

It is written line 124:

The miscuts were measured using the rocking curve method at a low angle (sample surface).

However, commonly,  in rocking curve technique, the detector is set at a specific Bragg angle.  So I assume, that futher reader can be confused as I was. Maybe reference can be provided?

3) The sentence in Lines 137-138 ends abruptly:

The samples with different anisotropies were examined and no correlation was observed between these anisotropies and the crystallographic.

That was meant?

4) In addition, Letters a) and b) in Fig.4 should be written with a large font size.

Reviewer 2 Report

Changes are appropriate and the manuscript can be accepted in its current form.

Author Response

Dear Reviewer

Thank you for all your insights.

Best regards

Reviewer 3 Report

The manuscript can be published in present form

Author Response

(The authors gave the same response as above.)
